# Protective Effect of Zeaxanthin from *Lycium barbarum* L. on Ultraviolet B-Induced Skin Photodamage in Mice Through Nrf2-Related Pathway

**DOI:** 10.3390/antiox14060632

**Published:** 2025-05-25

**Authors:** Lin Zhu, Qiruonan Shen, Yujuan Xu, Chunmei Li

**Affiliations:** 1College of Food Science and Technology, Huazhong Agricultural University, Wuhan 430070, China; zhu-lin@webmail.hzau.edu.cn (L.Z.); shenqiruonan@163.com (Q.S.); 2Sericultural & Agri-Food Research Institute, Guangdong Academy of Agricultural Sciences, Key Laboratory of Functional Products, Ministry of Agriculture, Guangdong Key Laboratory of Agricultural Products Processing, Guangzhou 510610, China; xyj6510@126.com

**Keywords:** UVB, skin, *Lycium barbarum* L., zeaxanthin, appearance, matrix metalloproteinases, tissue structure, Nrf2, antioxidant

## Abstract

Ultraviolet (UV) radiation is a predominant cause of skin damage, with UVB leading to more severe harm compared to UVA. *Lycium barbarum* L. (*L. barbarum*) is known for its high carotenoid content and has shown great potential in mitigating UVB-induced skin damage. This study investigated the protective effect and mechanism of zeaxanthin from *L. barbarum* on UVB-damaged skin in BALB/c mice. The results demonstrated that zeaxanthin effectively alleviated the UVB-injured appearance of mouse skin. Histological analyses revealed a reduction in epidermal thickness by 30% and 61% with low and high doses of zeaxanthin, respectively, compared to the model group. Zeaxanthin also inhibited the degeneration of elastic and collagen fibers. Further investigations indicated that the protective mechanism of zeaxanthin was not involved with inflammation suppression. Instead, it activated nuclear factor erythroid 2-related factor 2 (Nrf2) to approximately 3 times the level of the model group, significantly promoting the expression of various antioxidant enzymes and enhancing the total antioxidant capacity of skin tissue, subsequently reducing oxidative stress. Zeaxanthin also downregulated the expression of matrix metalloproteinases, reducing collagen degradation by 35% compared to the model group, which led to improved skin tissue structure and protection against UVB-induced photodamage. These findings provided a theoretical basis for the advanced development and high-value utilization of carotenoids in *L. barbarum*.

## 1. Introduction

Skin serves as a critical barrier between the internal and external environments of the human body, preventing the invasion of toxic substances and ultraviolet (UV) radiation. Among various external factors, UV light is a major contributor to skin damage, including skin cancers [1,2,3]. Acute UV exposure can result in adverse effects such as epidermal hyperplasia, telangiectasia, and increased collagen degradation. Prolonged exposure to high-intensity UV irradiation leads to skin relaxation, inelasticity, DNA damage accumulation, and, potentially, skin cancers [4,5,6]. Notably, UVB causes more severe skin damage compared to an equivalent dose of UVA radiation [7]. The directional absorption of UVB by DNA will lead to DNA damage and the formation of corresponding photodamage products [8]. UV radiation can cause the skin to produce a large amount of reactive oxygen species (ROS), leading to the excessive consumption of antioxidant enzymes and the disruption of the normal redox system, resulting in oxidative stress and damage to cells and tissues. Additionally, excessive ROS can activate various signaling pathways by regulating the expression of various receptors on the cell surface, triggering inflammatory reactions in the body. The interaction between oxidative stress and inflammatory response is considered the main cause of skin photodamage [9,10]. Therefore, active ingredients with antioxidant, anti-inflammatory, and UV absorption properties have excellent protective potential against skin damage caused by UV radiation. Meanwhile, the chemical components in sunscreen products may cause skin allergies and contact dermatitis, posing certain safety hazards. Therefore, finding natural ingredients with a potential preventative effect on skin damage caused by UVB radiation is of great significance.

Carotenoids, known for their antioxidant and anti-inflammatory activities, have shown a potential protective effect against photodamage [11,12,13]. And it has been found that the content of carotenoids in *Lycium barbarum* L. (*L. barbarum*) is higher than other common fruits and vegetables, with levels 3.39 times higher than tomatoes, 4.25 times higher than persimmons, and 5.36 times higher than carrots [14]. *L. barbarum* contains various important functional components, but current research mainly focuses on its polysaccharides. Therefore, the study of carotenoids in *L. barbarum* is essential. *L. barbarum* mainly contains three types of free carotenoids and three types of esterified carotenoids. Among them, the fully esterified form of zeaxanthin is the richest [15]. Zeaxanthin has multiple conjugated double bonds and terminal hydroxyl groups, which can effectively scavenge free radicals and have strong antioxidant properties with an IC_50_ value of 9.044 ppm for scavenging 1,1-diphenyl-2-pyridine hydrazide (DPPH) radicals [16]. Furthermore, it has been shown that zeaxanthin can inhibit oxidative stress in various tissues and cells, including the retina or liver [17,18,19,20], as well as conjunctival or retinal pigment epithelial cells [21,22]. In the skin, zeaxanthin can effectively increase hydration together with oil on the skin surface while significantly reducing lipid peroxidation levels [23]. Furthermore, zeaxanthin can protect the skin from H_2_O_2_-induced oxidative injury by inhibiting ROS production and preventing DNA damage [24]. Based on these studies and the gradually clarified connection between oxidative stress and UV-induced skin damage, we suppose that zeaxanthin may be expected to be applied in the protection against skin photodamage.

Therefore, this study aims to elucidate the role of zeaxanthin derived from *L. barbarum* in protecting against UVB-induced skin injury in vivo and explore its underlying mechanism. We established an acute UVB-induced skin injury model using BALB/c mice to observe the effect of zeaxanthin on skin appearance and tissue structure. Additionally, we detected collagen content, matrix metalloproteinase (MMP) levels, total antioxidant capacity, antioxidant enzyme activities, inflammatory factor levels, and nuclear factor erythroid 2-related factor 2 (Nrf2) expression to investigate the protective effects and mechanisms of zeaxanthin on UVB-injured skin.

## 2. Materials and Methods

### 2.1. Mice

All experiments were approved by the Laboratory Research Center and Ethics Committee of Huazhong Agricultural University (HZAUMO-2021-0177; 19 November 2021). Forty female BALB/c mice (SPF grade), weighing 18.0 ± 2 g, were housed at an environmental temperature of 22–24 °C and a relative humidity of 55% ± 10% with a light/dark cycle for 12 h. All were ensured free access to a diet.

### 2.2. Preparation of Zeaxanthin

Zeaxanthin was obtained from L. barbarum following the method established by Shen et al. [25]. Briefly, the L. barbarum (Ningxiahong Goji Industry Group Co., Ltd., Ningxia, China) was freeze-dried for 48 h, crushed by a crusher, and then passed through a 60-mesh screen to obtain a dry powder. A certain amount of dry powder was taken, and 6.3% KOH (Sinopharm Chemical Reagent Co., Ltd., Shanghai, China) (g/g) was added. Then, an ethanol solution of 9% 1-hexyl-3-methylimidazole acetate (Shanghai Cheng Jie Chemical Co., Ltd., Shanghai, China) (g/mL) was added at a solid–liquid ratio of 1:40 (g/mL), and the extraction was carried out under ultrasonic conditions at 420 W for 39 min. After centrifuging for 5 min at 9500× *g* at 4 °C, the extract was collected and extracted multiple times with methyl tert-butyl ether and saturated sodium chloride solution until the lower layer was colorless. The supernatant was combined and washed with water until neutral. Then, zeaxanthin was obtained by concentrating and evaporating it until dry at 35 °C in a rotary evaporator (Shanghai Yarong Biochemical Instrument Factory, Shanghai, China).

### 2.3. Experimental Design

Mice were depilated on the back (4 cm × 2 cm) two days prior to the experiment and randomly divided into four groups. Thirty minutes before UVB irradiation, the mice’s skin in the administration groups was externally treated daily with 0.12 mL of low-dose (0.045 mg/mL, Z-L) or high-dose (0.45 mg/mL, Z-H) zeaxanthin, using Ya et al.’s study as a reference [26]. The model group (MC) received the same amount of solvent (propylene glycol/ethanol = 8:2). Then, they were all exposed to UVB irradiation (280–315 nm) at a vertical distance of 8 cm, receiving a daily irradiation dose of 240 mJ/cm^2^ for six consecutive days until obvious redness, peeling, pigmentation, and festers appeared in the MC group. The normal control group (NC) was neither treated nor exposed to light. After the experiment, mice were euthanized by cervical dislocation, and skin tissues were collected 24 h after the final UVB irradiation. The tissues were visually inspected and fixed in 4% paraformaldehyde (Servicebio, G1101, Wuhan, China) for histological analyses, or preserved in liquid nitrogen and stored at an ultra-low temperature for further use.

### 2.4. Histological Analyses

Skin tissues were fixed in 4% paraformaldehyde (Servicebio, G1101) for over 24 h and embedded in paraffin. Then, the tissues were sliced into sections of 4 μm thickness and stained with hematoxylin and eosin (HE), Masson, and resorcinol fuchsin (Servicebio, HE, G1005; Masson, G1006; resorcinol fuchsin, G1054). The sections were observed using optical microscopes (Nikon, Tokyo, Japan) and photographed using a digital camera (Nikon, Tokyo, Japan). For each stained section, 3–5 visual fields were randomly selected, and 3–5 different parts were analyzed within each field. Epidermis (ED) thickness was measured using HE-stained sections. Dermis collagen fibers (CFs) and elastic fibers (EFs) were examined by Masson and resorcinol fuchsin staining, respectively [27,28].

### 2.5. Assays and Kits

Hydroxyproline (Hyp) is one of the characteristic amino acids of collagen, accounting for 13.4% of it [29]. Collagen content in skin tissues was quantified by determining Hyp content with kits from the Nanjing Jiancheng Bioengineering Institute (A030-2-1, Nanjing, China). Total antioxidant capacity (T-AOC) as well as superoxide dismutase (SOD), catalase (CAT), and glutathione peroxidase (GSH-Px) activities in skin tissues were determined with kits from the Nanjing Jiancheng Bioengineering Institute (T-AO, A015-1-2; SOD, A001-1-1; CAT, A007-1-1; GSH-Px, A005-1-2). Interleukin (IL)-1β and IL-6 in skin tissues were quantified using ELISA kits from Shanghai Huding Biotechnology Co., Ltd. (Shanghai, China, IL-1β, D-20532; IL-6, D-20012).

### 2.6. Immunofluorescence Staining

Skin tissues were fixed in 4% paraformaldehyde (Servicebio, G1101) for over 24 h and embedded in paraffin. Then, the tissues were sliced into sections of 4 μm thickness. Paraffin sections were dewaxed and subjected to antigen retrieval using EDTA buffer (pH 8.0, Servicebio, G1206) in a microwave oven. After the sections were slightly dried, a circle was drawn around the tissue with a histochemical pen, and then BSA (Servicebio, GC305010) was added to the circle and incubated for 30 min. After incubating with the primary antibody (Servicebio, MMP-3, GB11131; MMP-9, GB11132) overnight at 4 °C, sections were washed with PBS (Servicebio, G0002) and incubated with the secondary antibody (Servicebio, HRP-conjugated Goat Anti-Rabbit IgG (H+L), G1213). Then, the cell nuclei were re-stained with DAPI (Servicebio, G1012). After autofluorescence quenching, the tablets were sealed with anti-fluorescence quenching mounting agents (Servicebio, G1401). Images were captured using optical microscopes and photographed using a digital camera (Nikon, Tokyo, Japan).

### 2.7. Western Blotting

Total proteins of skin tissues were extracted using ice-cold RIPA buffer (Servicebio, G2002) supplemented with a phosphatase and protease inhibitor cocktail (Servicebio, G2007, G2006). Protein samples were separated by SDS-PAGE gel, then transferred to PVDF membranes (Millipore, IPVH00010, Shanghai, China), and blocked for 1 h at room temperature with 5% skimmed milk powder (Servicebio, G5002). The membranes were incubated with primary antibodies (Nrf2, Proteintech Group, 16396-1-ap, Wuhan, China; β-actin, Servicebio, GB15001) overnight at 4 °C, and then incubated with secondary antibodies (HRP-conjugated Goat Anti-Rabbit IgG (H+L), Servicebio, GB23303) at room temperature for 30 min. The blots were visualized using chemiluminescent blotting reagents and imaged in the imaging system (EPSON, Suwa, Japan). Quantification was performed using grayscale analysis software (AlphaEaseFC 4.0). β-actin was used as a loading control for the total protein.

### 2.8. Statistical Analysis

Data were expressed as mean ± SEM. GraphPad Prism 8.0.2 software was used for image rendering. The results were analyzed in terms of variance (ANOVA) using SPSS 19.0. Statistical difference was set at *p* < 0.05.

## 3. Results and Discussion

### 3.1. L. barbarum Zeaxanthin Alleviated UVB-Damaged Skin Appearance of Mice

As depicted in Figure 1A, the back skin of the NC group exhibited fine texture and good elasticity, without any swelling or obvious wrinkles. However, the mice of the MC group after continuous UVB irradiation showed inflamed skin with obvious eschar, swelling, festering, and pigmentation. In contrast, zeaxanthin treatment alleviated these damages, as evidenced by slightly improved swelling and eschar in the Z-L group, and significantly reduced skin petechiae, erythema, and eschar in the Z-H group compared to the MC group. Figure 1B illustrates the changes in subcutaneous blood vessels. The mice of the NC group were light pink with no obvious pigmentation, petechiae, or ecchymosis. However, the subcutaneous blood vessels in the MC group were significantly expanded after UVB irradiation. Dark purple blood vessels and numerous ecchymoses and petechiae were also observed. Zeaxanthin treatment improved this damage, with the Z-L group showing alleviated expansion, ecchymoses, and pigmentation. But there were still significant differences compared to the NC group. In contrast, the Z-H group showed the most evident improvement, with slight congestion, less pigmentation, and light red vessels, closely resembling the NC group. This aligns with Ya et al. [26], who demonstrated the inhibitory effect of lavender flavonoids on adverse pathological changes in subcutaneous vessels.

These findings suggested that zeaxanthin exerted a protective effect on the appearance of UVB-injured skin.

### 3.2. L. barbarum Zeaxanthin Improved UVB-Injured Skin Tissue Structure of Mice

We then examined the effect of *L. barbarum* zeaxanthin on the skin tissue structure via histological analyses.

HE staining results (Figure 2A) showed that the NC group had a thin ED and stratum corneum (SC), and the ED was tightly combined with the dermis (DR). Additionally, the dermal–epidermal junction (DEJ) was in regular waves, and normal hair follicles (HFs) could be observed. After continuous UVB irradiation, the tissue structure in the MC group changed significantly, and its arrangement became disordered. The ED and SC were significantly thickened. Disappearing DEJs and keratinized HFs were also observed in the MC group. In contrast, zeaxanthin treatment notably decreased the thickness of ED and SC compared with the MC group. The DEJs in Z-L and Z-H groups were gradually visible, and normal HFs could also be observed. It is worth noting that the tissue structure in the Z-H group closely resembled that of the NC group. UVB irradiation dramatically thickened the ED of mice by about 5 times compared to the NC group (68.40 µm vs. 13.49 µm, Figure 2D), while Z-L and Z-H treatment reversed the increase by 30% and 61%, respectively (*p* < 0.05), illustrating that zeaxanthin could effectively mitigate UVB-induced adverse changes in skin tissues. These results were similar to those of Zhong et al. [30], who reported the photoprotective effect of andrographolide sodium bisulfate on the skin of UV-injured mice.

EFs and CFs are essential for skin tissue structure, and they can effectively maintain the toughness and elasticity of the skin [31,32]. Next, the changes in the EFs and CFs of mouse skin were observed by resorcinol fuchsin and Masson staining, respectively. The changes in skin EFs are shown in Figure 2B. The NC group had wavy, slender, and orderly arranged EFs around the CFs, while UVB irradiation in the MC group caused serious EF degeneration and irregular distribution. Some EFs were entangled and denatured with each other, which is similar to the phenomenon observed by Jiatong et al. [33] in UV-injured mouse skin. Compared with the MC group, this phenomenon in Z-L and Z-H groups was alleviated in both, and relatively well-distributed EFs with significantly reduced fractures and wavy, slender shapes could be observed in the Z-H group. This aligned with the results of Xu et al. [34], who found that tetrahydrocurcumin had a protective effect against UV-injured mouse skin. The changes in skin CFs are shown in Figure 2C. The CFs of the NC group were evenly distributed, tightly arranged, darker colored, and parallel to the ED. After UVB irradiation, the CFs in the MC group were sparse and not arranged closely compared with the NC group, and the color stained by Masson became significantly lighter, indicating the decreased content of collagen. Adverse phenomena such as curling and breaking could also be noticed, which is similar to that observed by Khan et al. [28] in UV-injured mouse skin. After treatment with zeaxanthin, the CFs of both Z-L and Z-H groups were in an orderly arrangement and parallel to the ED. The curling and breaking were obviously reduced. CFs were mainly composed of collagen; thus, the content of collagen was subsequently determined to further explore the changes in skin CFs. It can be seen from Figure 2E that the collagen content in the MC group was decreased significantly after UVB irradiation compared with the NC group, which is consistent with the results of Zhi et al. [35]. Z-L treatment slightly increased collagen content without a significant difference, while Z-H treatment increased it by 35% compared to the MC group (*p* < 0.05). These results were consistent with Masson staining, as shown in Figure 2C, suggesting that zeaxanthin might promote the synthesis or reduce the degradation of collagen in the skin, thereby protecting CFs.

UV irradiation can enhance the expression of MMP-3 (matrix protease) and MMP-9 (gelatinase) in dermal fibroblasts and keratinocytes, leading to CF degradation and EF synthesis disorders [36,37]. Therefore, the expression of MMP-3 and MMP-9 was detected to further explore the effect of *L. barbarum* zeaxanthin on UVB-injured mouse skin tissues, and the results are shown in Figure 3. The expression of MMP-3 (Figure 4A) and MMP-9 (Figure 4B) in the MC group was increased significantly after UVB irradiation, while Z-L and Z-H treatment both decreased them obviously. The results were consistent with the changes in the EFs and CFs observed by resorcinol fuchsin and Masson staining above. Shin et al. [38] and Tominaga et al. [39] also found that carotenoids or astaxanthin could alleviate UVB-induced photoaging via downregulating MMPs, which is consistent with our results.

### 3.3. L. barbarum Zeaxanthin Inhibited UVB-Induced Oxidative Stress of Mouse Skin via Activating Nrf2

Given that MMP production is related to UV-induced oxidative stress and inflammation [9], we hypothesized that zeaxanthin might improve UVB-injured skin through antioxidant or anti-inflammatory pathways. Unfortunately, we found that zeaxanthin treatment showed a limited effect on improving the inflammatory state of UVB-injured mouse skin, which was evidenced by the slight but not significant decrease in interleukin (IL)-1β and IL-6 content compared with the MC group (Appendix A). Thus, we subsequently focused on the effect of zeaxanthin on the UVB-induced oxidative stress of mouse skin.

We first measured the levels of T-AOC, SOD, CAT, and GSH-Px in the skin of mice, which revealed significant decreases in the MC group compared to the NC group (Figure 4). In contrast, zeaxanthin treatment evidently reversed the phenomena. CAT activities in the Z-L and Z-H groups were increased by approximately 33% and 66%, respectively (Figure 4C, *p* < 0.05). GSH-Px activities were enhanced by about 26% and 32%, respectively (Figure 4D, *p* < 0.05). Z-L treatment did not show an improving effect on T-AOC and SOD levels, while Z-H treatment noticeably increased these levels by 39% and 18%, respectively (Figure 4A,B). These results illustrate the role of zeaxanthin in inhibiting the UVB-induced oxidative stress on mouse skin. Our findings were consistent with that of Wang et al. [40], who reported similar antioxidant enzyme activity increases with pogostone.

Nrf2, a key factor in antioxidant pathways [41,42], plays an important role in UV-injured mouse skin [43,44]. Thus, Nrf2 was further analyzed to investigate whether zeaxanthin could inhibit UVB-induced oxidative stress via activating Nrf2. As shown in Figure 5, Nrf2 expression in the MC group was decreased significantly compared with the NC group, but zeaxanthin treatment increased it by approximately 3 times (*p* < 0.05), with no significant difference between Z-L and Z-H groups. This indicated that *L. barbarum* zeaxanthin could inhibit UVB-induced oxidative stress via Nrf2 activation, thereby improving skin injury.

Previous studies have shown that due to the different structural properties of carotenoids, they can protect against UVB-induced damage through various mechanisms (Table 1). For instance, lycopene has been shown to inhibit tumorigenesis-related epidermal ornithine decarboxylase activity and downregulate the expression of enzymes related to the apoptosis pathway [45], while β-carotene enhances Nrf2 and heme oxygenase (HO)-1 expression, thereby reducing oxidative stress [46]. Fucoxanthin has been shown to reduce cyclooxygenase (COX)-2 or enhance Nrf2 and HO-1 expression, thereby inhibiting inflammation, which was not observed in β-carotene [13,46]. This study found that zeaxanthin could activate related signaling pathways by promoting Nrf2 expression (about 3 times that of the model group), increasing the various antioxidant enzyme contents, and improving the T-AOC level of skin tissue, thereby enhancing the antioxidant defense ability of the body. Additionally, it inhibited the expression of MMPs, reducing the breakdown of collagen, thereby effectively improving the skin tissue structure and protecting it against UVB irradiation-induced skin damage in mice. Meanwhile, this study found that zeaxanthin treatment did not significantly reduce the levels of the two inflammatory factors, IL-1β and IL-6, which might be due to its limited anti-inflammatory activity in UVB-damaged mouse skin. Therefore, our findings indicated that zeaxanthin predominantly exerted its protective effect on UVB-injured skin through antioxidant pathways rather than anti-inflammatory pathways. It is worth noting that the chemical components in sunscreen products may cause skin allergies and contact dermatitis, posing certain safety hazards. In contrast, as a natural compound, the topical application of zeaxanthin in protecting against skin photodamage is safer.

UV irradiation that is acceptable to the human body includes UVA and UVB, generally in low-dose and long-term forms. Therefore, future studies should adopt UVA and UVB irradiation combinations to establish a low-dose and long-term photoaging model, so as to explore the protective effect of zeaxanthin under this condition. More comprehensive studies on the protective mechanism of zeaxanthin are also needed.

## 4. Conclusions

UVB irradiation could cause significant damage to the appearance and tissue structure of skin, whereas *L. barbarum* zeaxanthin had a protective effect against UVB-induced skin damage. Specifically, zeaxanthin effectively mitigated UVB-induced skin swelling, eschar pigmentation, and subcutaneous ecchymosis. It also protected EFs and CFs from degeneration. The protective mechanism of zeaxanthin against UVB damage was through Nrf2 activation, promoting the expression of antioxidant enzymes (SOD, CAT, and GSH-Px) and enhancing the T-AOC of skin tissue, subsequently reducing oxidative stress in the skin. Additionally, zeaxanthin downregulated the expression of MMP-3 and MMP-9, thereby reducing collagen degradation and improving skin tissue structure. Our data indicated that the photoprotective role of zeaxanthin was primarily through regulating antioxidant rather than anti-inflammatory pathways. These findings suggest that *L. barbarum* zeaxanthin holds potential for applications in skin protection products.

## Figures and Tables

**Figure 1 antioxidants-14-00632-f001:**
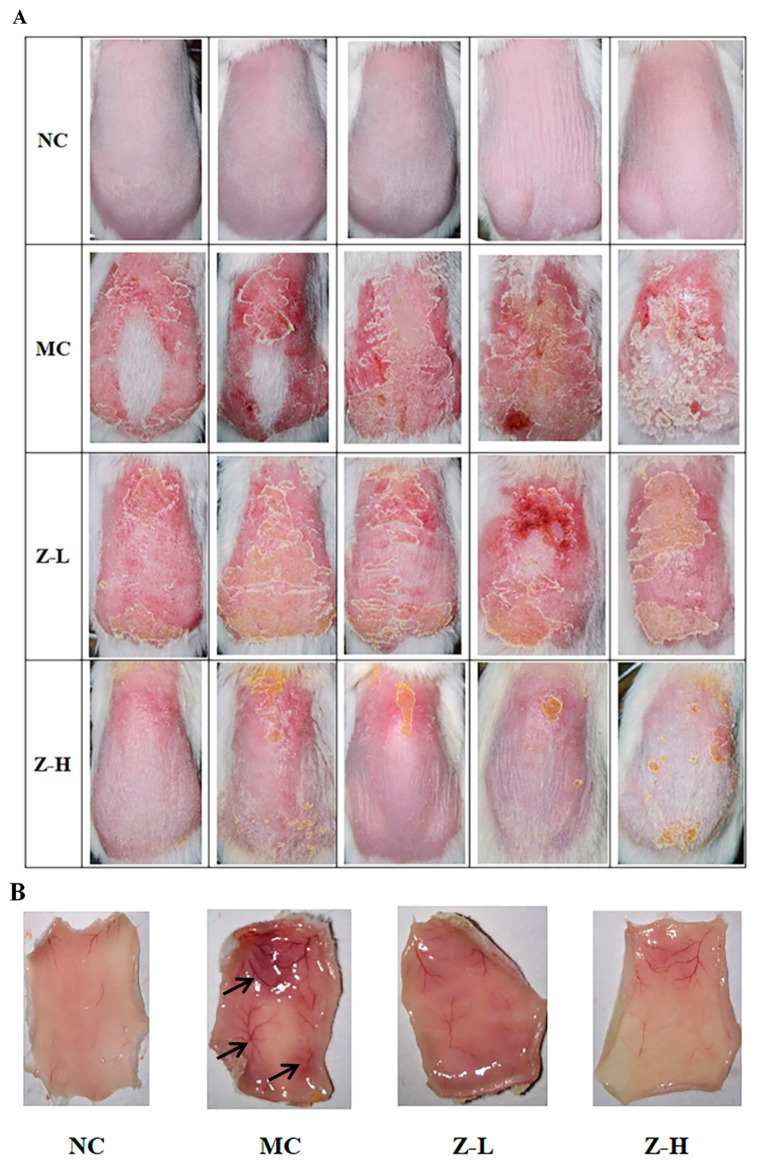
(**A**) Appearance of the back skin of mice at the end of the experiment. (**B**) Changes in the subcutaneous blood vessels of the skin. Black arrows in MC indicate degenerated subcutaneous blood vessels (*n* = 5).

**Figure 2 antioxidants-14-00632-f002:**
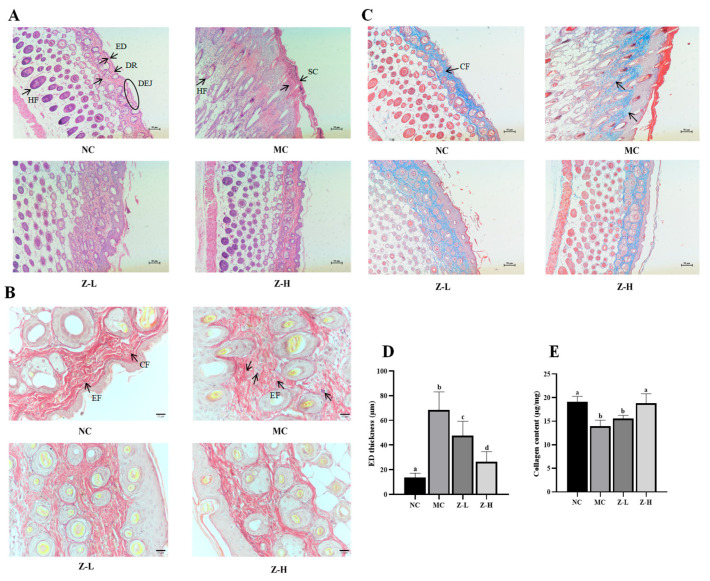
(**A**) HE staining results (100×) of mouse skin. Black arrows indicate the layers of organization. (**B**) Resorcinol fuchsin staining results (400×) of EFs. Black arrows in NC indicate EFs and CFs, respectively. Black arrows in MC indicate degenerated EFs. (**C**) Masson staining results (100×) of CFs. Black arrows in NC indicate CFs. (**D**) Statistical results of ED thickness. (**E**) Collagen content in skin tissues. ED, epidermis; DR, dermis; DEJ, dermal–epidermal junction; HF, hair follicle; SC, stratum corneum; CF, dermis collagen fibers; EF, elastic fibers. Different letters indicate significant differences at *p* < 0.05. Data are expressed as mean ± SEM (*n* = 3–4).

**Figure 3 antioxidants-14-00632-f003:**
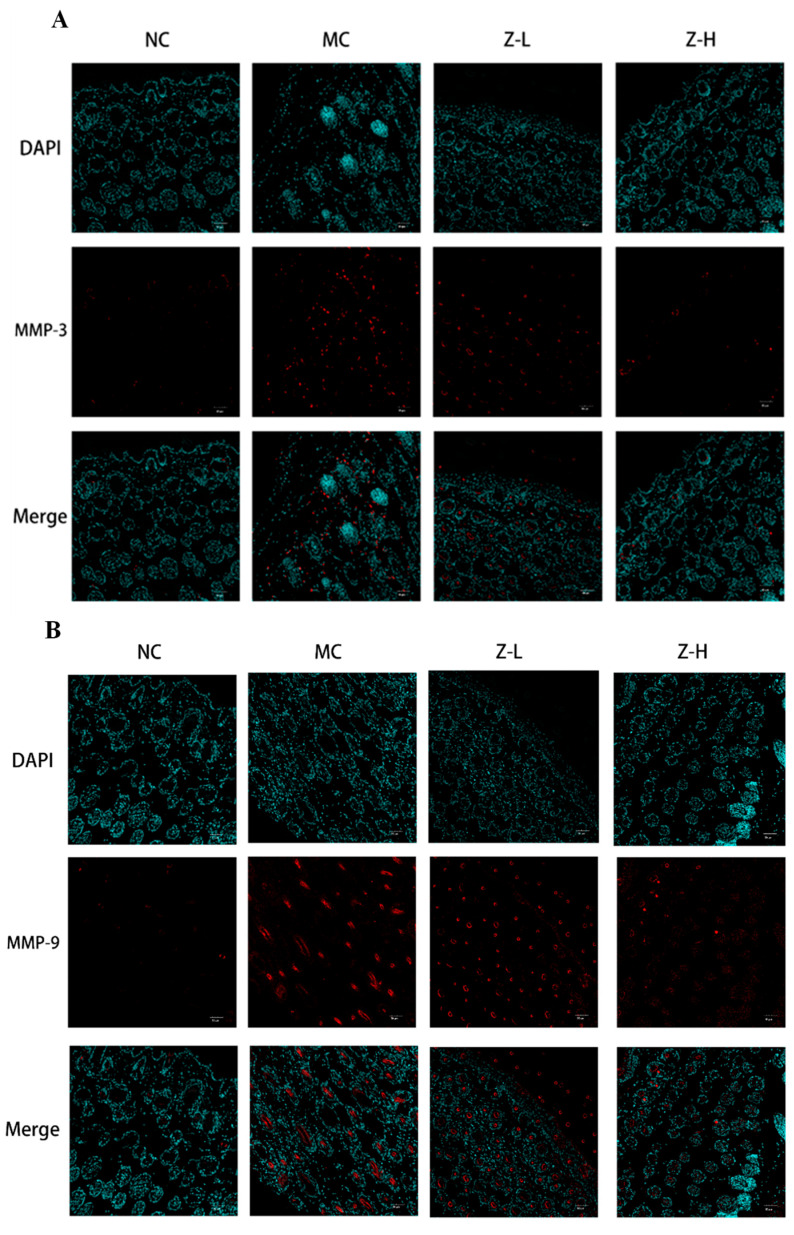
Immunofluorescence detection of MMP-3 (**A**) and MMP-9 (**B**) expression in the skin tissues of mice (200×) (*n* = 3–5).

**Figure 4 antioxidants-14-00632-f004:**
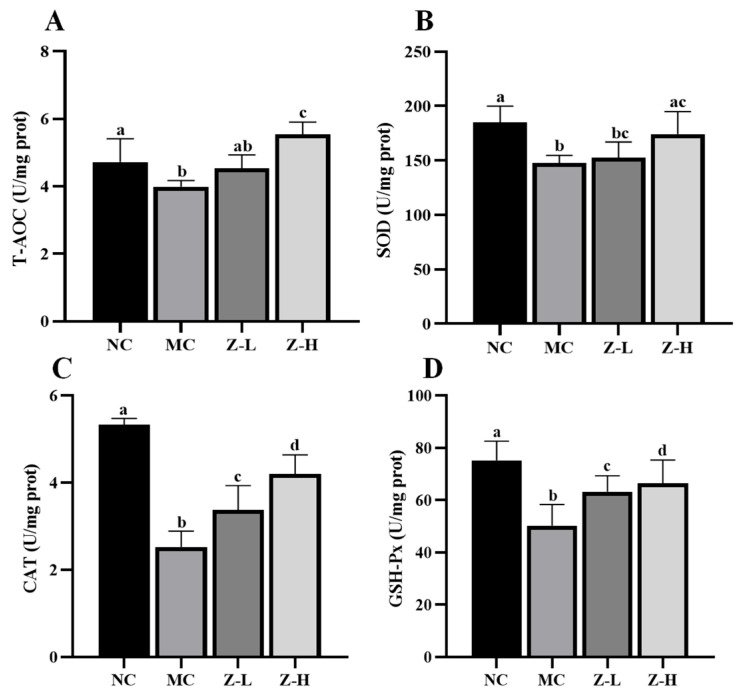
T-AOC and antioxidant enzyme levels in the skin tissues of mice: (**A**) T-AOC; (**B**) SOD; (**C**) CAT; (**D**) GSH-Px. Different letters indicated significant differences at *p* < 0.05. Data are expressed as mean ± SEM (*n* = 5–6).

**Figure 5 antioxidants-14-00632-f005:**
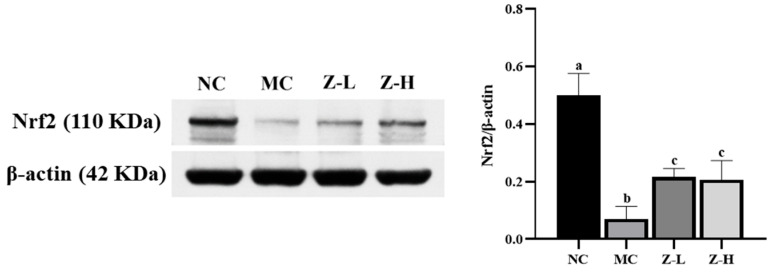
Expression of Nrf2 protein in the skin tissues of mice (Nrf2, 110 KDa; β-actin, 42 KDa). Different letters indicated significant differences at *p* < 0.05. Data are expressed as mean ± SEM (*n* = 3).

**Table 1 antioxidants-14-00632-t001:** Protective effects and mechanisms of zeaxanthin and other carotenoids on UVB-damaged mouse skin.

Carotenoids	Protective Effect of Carotenoids on UVB-Damaged Mouse Skin
lycopene	reduces epidermal ornithine decarboxylase and caspase-3; inhibits tumor and apoptosis
β-carotene	enhances Nrf2 and HO-1; inhibits oxidative stress
fucoxanthin	enhances Nrf2 and HO-1; reduces COX-2; inhibits oxidative stress and inflammation
zeaxanthin	enhances Nrf2; reduces MMPs; inhibits oxidative stress

## Data Availability

Data will be made available upon request.

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
