# Peer review of "Protective Effect of Zeaxanthin from Lycium barbarum L. on Ultraviolet B-Induced Skin Photodamage in Mice Through Nrf2-Related Pathway"

_antioxidants, 2025, doi:10.3390/antiox14060632_

Round 1

Reviewer 1 Report

The paper by Zhu et al. Discussed the protective effect of zeaxanthin of Lycium barbarum origin against UVB-induced skin photodamage in mice and its mechanism. The study design was reasonable and the data provide some insight into a theoretical basis for the application of zeaxanthin in skin photoprotection.

Yet, the authors should consider the following modifications for improvement of this manuscript:

Major points:

  1. The authors proposed that zeaxanthin acts through the Nrf2 pathway, but key downstream targets (e.g. HO-1, NQO1) were not tested. Elevated Nrf2 protein expression alone is not a very strong evidence to prove pathway activation. Besides, the authors should also consider testing other antioxidant pathways or protective mechanisms.
  2. The article states that zeaxanthin may not act through the anti-inflammatory pathway, but without detecting more inflammatory markers (e.g., TNF-α, COX-2), it is too lopsided to draw a conclusion based on the results of IL-1β and IL-6 alone. And the results of ELISA for inflammatory factors (IL-1β, IL-6) did not show significant differences, the author should discuss possible reasons.
  3. The similarities and differences between zeaxanthin and other carotenoids (e.g., β-carotene, lutein) in terms of their photoprotective effects can be added to highlight their uniqueness. Ideally, the author should consider adding a figure or table describe the differences.

Minor points:

  1. For Fig. 2 ED Thickness and collagen content, the authors should explicit biological replicates and provide raw data.

2.The basis for UVB irradiation dose (240 mJ/cm²) and frequency (6 consecutive days) was not cited from the literature or pre-experimental data.

  1. An evaluation of the safety of topical application of zeaxanthin could be added to the discussion section to support its potential application value.

Overall, the paper offers some insight into the skin benefits of zeaxanthin, but at this moment, the data is still not sufficient enough to support all the conclusions. Addressing the issues mentioned above could hopefully further enhance its quality and impact. I recommend reconsideration after major revision.

Ideally, a statistician could help to double check the results, but it might not be necessary.

Reviewer 2 Report

The study analysed the protecting effect of zeaxanthin obtained from Lycium barbarum L. on skin injury in mice during a feeding trial. The study is well presented and justified. Some performances or clarifications could be done before it is accepted for publication.

Abstract

Line 18: Further investigations ? Does it correspond to the current study ?

Line 26: Can the results be generalised to carotenoids ?

Keywords

Include feeding trial and Lycium babarum L.

Material and methods

Line 95: … 9500xg …

A wide range of complementary and valuable analyses were carried out regarding the effect of zeaxanthin in mice.

Lines 104-105: Provide some justification regarding the low and high doses that were tested in the experiment.

Lines 161-164: Were there replicates carried out ?

Conclusions

After proving the positive effect, an on-coming research that could be expressed in this section is the optimisation of the feeding conditions, i.e., the zeaxanthin concentration provided during the feeding trial.
